# Cooperative interactions between seed-borne bacterial and air-borne fungal pathogens on rice

Boknam Jung[1], Jungwook Park[2], Namgyu Kim[2], Taiying Li[1], Soyeon Kim[1], Laura E. Bartley [3], Jinnyun Kim[2], Inyoung Kim[2], Yoonhee Kang[1], Kihoon Yun[1], Younghae Choi[1], Hyun-Hee Lee[2], Sungyeon Ji[1], Kwang Sik Lee[1], Bo Yeon Kim[1], Jong Cheol Shon[4], Won Cheol Kim[4], Kwang-Hyeon Liu[4], Dahye Yoon[5], Suhkman Kim[5], Young-Su Seo[2] & Jungkwan Lee[1]

Bacterial-fungal interactions are widely found in distinct environments and contribute to ecosystem processes. Previous studies of these interactions have mostly been performed in soil, and only limited studies of aerial plant tissues have been conducted. Here we show that a seed-borne plant pathogenic bacterium, *Burkholderia glumae* (Bg), and an air-borne plant pathogenic fungus, *Fusarium graminearum* (Fg), interact to promote bacterial survival, bacterial and fungal dispersal, and disease progression on rice plants, despite the production of antifungal toxoflavin by Bg. We perform assays of toxoflavin sensitivity, RNA-seq analyses, lipid staining and measures of triacylglyceride content to show that triacylglycerides containing linolenic acid mediate resistance to reactive oxygen species that are generated in response to toxoflavin in Fg. As a result, Bg is able to physically attach to Fg to achieve rapid and expansive dispersal to enhance disease severity.

[1] Department of Applied Biology, Dong-A University, Busan 49315, Korea. [2] Department of Microbiology, Pusan National University, Busan 46269, Korea. [3] Department of Microbiology and Plant Biology, University of Oklahoma, Norman, OK 73019, USA. [4] BK21 Plus KNU Multi-Omics-Based Creative Drug Research Team, College of Pharmacy and Research Institute of Pharmaceutical Sciences, Kyungpook National University, Daegu 41566, Korea. [5] Department of Chemistry and Chemistry Institute for Functional Materials, Pusan National University, Busan 46269, Korea. Correspondence and requests for materials should be addressed to Y.-S.S. (email: yseo2011@pusan.ac.kr) or to J.L. (email: jungle@dau.ac.kr)

Bacterial-fungal interactions (BFI) widely exist in distinct environments, such as on the human body, in food and in soil[1–4]. These interactions often have a physical or molecular basis[2,5], and they can lead to numerous biological effects that vary from antagonism to cooperation[2,6–8]. Some of these effects involve metabolite exchange, signaling chemotaxis and physical communication, such as planktonic form, mixed biofilm and intra-hyphal colonisation[1]. Symbiotic relationships constituted in particular niches can also result in the translocation of nutrients, resistance to environmental stress and pathogenic acquisition[6,7].

Beneficial effects of BFI have previously been described for *Burkholderia terrae* and *Lyophyllum* sp. in soil[7]. For example, the presence of *Lyophyllum* sp. has been found to be essential for the colonisation and transportation of *Burkholderia terrae* in soil[9], and the bacteria exhibited better survival in acidic soils with fungal exudates as nutrients[10,11]. Similarly, *Burkholderia rhizox-inica* lives within fungal hyphae and is crucial for fungal pathogenicity toward rice seedlings in soil[12,13]. To date, molecular and physical interactions between *Burkholderia* and fungi have been well studied in soil, while only limited studies of BFIs between bacteria and air-borne fungi have been conducted.

The seed-borne bacterial plant pathogen, *Burkholderia glumae* (Bg), is one of the causal agents of bacterial panicle blight in rice fields[14]. This bacterium produces toxoflavin, which is both a critical virulence factor and an antimicrobial that induces production of superoxides and hydrogen peroxide ($H_2O_2$)[14,15]. Thus, Bg has the potential to monopolise rice grains by blocking the growth of saprophytic fungi and other pathogens[16,17]. However, we have observed that the air-borne fungal plant pathogen, *Fusarium graminearum* (Fg), is resistant to toxoflavin[18].

Here we examined the potential for Bg and Fg to interact and co-exist, particularly regarding the mechanism by which Fg develops resistance to toxoflavin that is produced by Bg. This study was facilitated by the generation of toxoflavin-sensitive Fg mutant strains and experiments performed in planta. The data obtained demonstrate that the two pathogens cooperatively interact, and this provides Fg with an opportunity to produce more spores and toxins and it provides Bg with an opportunity to achieve aerial dispersal.

## Results

**Bg is often isolated with Fg from field-grown rice grains.** Rice grains were collected from black- and glutinous-type rice crops at intervals of 7 days (d) from the flowering times. Both Fg and Bg were frequently observed on the collected grains (Fig. 1a) and their coexistence frequencies were quantitated with a PCR-based assay that identified Fg and Bg as 420 bp and 138 bp products, respectively, (Fig. 1b). Sequencing of the PCR products was performed to confirm the detection of Bg and Fg (data not shown). Among the field-grown grains examined, 40% (range: 22–52%) carried Bg alone, 1% (range: 0–4%) carried Fg alone, 24% (range: 2–36%) carried both pathogens, and 35% (range: 8–56%) were pathogen-free (Fig. 1c). Thus, Bg did not monopolise the rice grains examined and Fg was able to cohabitate with Bg on the rice grains.

**Isolation of Fg mutant strains sensitive to toxoflavin.** To understand the mechanism by which Fg is resistant to toxoflavin, a transcription factor deletion mutant library of the wild-type (WT) Fg strain, GZ03639, was screened. Three toxoflavin-sensitive mutants, *ΔGzZC190*, *ΔGzbZIP005* and *ΔGzC2H008* lacking *FGSG_07589*, *FGSG_02939* and *FGSG_01106* genes, respectively, were identified. *ΔGzZC190* exhibited the highest sensitivity to toxoflavin, followed by *ΔGzbZIP005* and *ΔGzC2H008* (Fig. 2a, Supplementary Fig. 1). All three mutants

and the WT strain were evaluated for 17 phenotypes, including vegetative growth, sexual development, conidiation, virulence, toxin production and stress responses, among others. These phenotypes remained unchanged in the *ΔGzZC190* and *ΔGzbZIP005* strains, except for sensitivity to toxoflavin. In contrast, *ΔGzC2H008* exhibited pleiotropic phenotypic changes, including abnormal sexual development, loss of virulence, and reduced toxin production, consistent with previous results (http://ftfd.snu.ac.kr/FgTFPD)[19].

The TF genes, *FGSG_07589*, *FGSG_02939*, and *FGSG_01106*, contain $Zn(II)_2Cys_6$, bZIP, and $C_2H_2$ zinc finger DNA-binding domains, respectively. *FGSG_07589* and *FGSG_02939* are less conserved compared to *FGSG_01106* among ascomycete fungi, and *FGSG_07589* is more variable than *FGSG_02939* in the *Fusarium* genus (Supplementary Fig. 2). *FGSG_01106* is a homologue of arsenite resistance protein 2 (ARS2) and is required for a DNA damage response[20], while the functions of the other two genes have not been studied. We mainly focused on *FGSG_07589* because it is more unique to Fg and its deletion resulted in greater sensitivity to toxoflavin.

Vegetative growth of *ΔGzZC190* and the WT strain were directly compared. The growth of *ΔGzZC190* was normal in the absence of toxoflavin and was dramatically reduced in toxoflavin-supplemented media (Fig. 2a). Under the same conditions, growth of the WT strain was diminished, yet still robust (Fig. 2a). To confirm the specificity of *FGSG_07589* to a toxoflavin response, *FGSG_07589* was amplified from the WT strain and reintroduced into *ΔGzZC190* to generate a *GzZC190*-complemented strain (GzZC190c) (Supplementary Fig. 3). Toxoflavin sensitivity was subsequently restored in GzZC190c (Fig. 2). Furthermore, in minimal medium (MM) containing toxoflavin, *ΔGzZC190* exhibited delayed spore germination, while normal spore germination was observed in the WT and GzZC190c strains (Supplementary Fig. 4).

**Transcriptome analyses of Fg strains in response to toxoflavin.** RNA-seq analyses were performed to profile the gene expression patterns of WT, *ΔGzZC190*, *ΔGzbZIP005*, and *ΔGzC2H008* strains to toxoflavin. First, we monitored the genes of *FGSG_07589*, *FGSG_02939*, and *FGSG_01106* in the WT strain. Only *FGSG_07589* was found to be highly induced by exposure to toxoflavin (Supplementary Table 1). With a ≥twofold change in expression selected as the criteria, 1062 genes were upregulated and 1216 genes were downregulated in the WT strain compared to *ΔGzZC190* ($P < 0.05$). According to the same criteria, 579 genes were upregulated and 881 genes were downregulated in the WT strain compared to *ΔGzbZIPH005*; and 1064 genes were upregulated and 1053 genes were downregulated in the WT strain compared to *ΔGzC2H008*. For the differentially expressed genes (DEGs), a Kyoto Encyclopedia of Genes and Genomes (KEGG) pathway-enrichment was performed. This analysis (with $P < 0.05$ and ≥5 gene count numbers) revealed that ~22% of pathways in the WT strain were upregulated compared to *ΔGzZC190* (Supplementary Table 2), while ~75% of pathways in the WT strain were upregulated compared to the other two mutants (Supplementary Tables 3 and 4). These results suggest that a greater number of pathways were induced in *ΔGzZC190* in response to toxoflavin compared to the WT strain. In particular, glycolysis/gluconeogenesis pathways were upregulated, while alanine, aspartate and glutamate metabolism pathways were downregulated, in the WT strain compared to all three mutants. In contrast, diverse metabolic pathways, including amino acid, carbon, and lipid metabolism pathways, were very dynamic in all three mutants. Furthermore, among the pathways related to protection from reactive oxygen species, which is the main

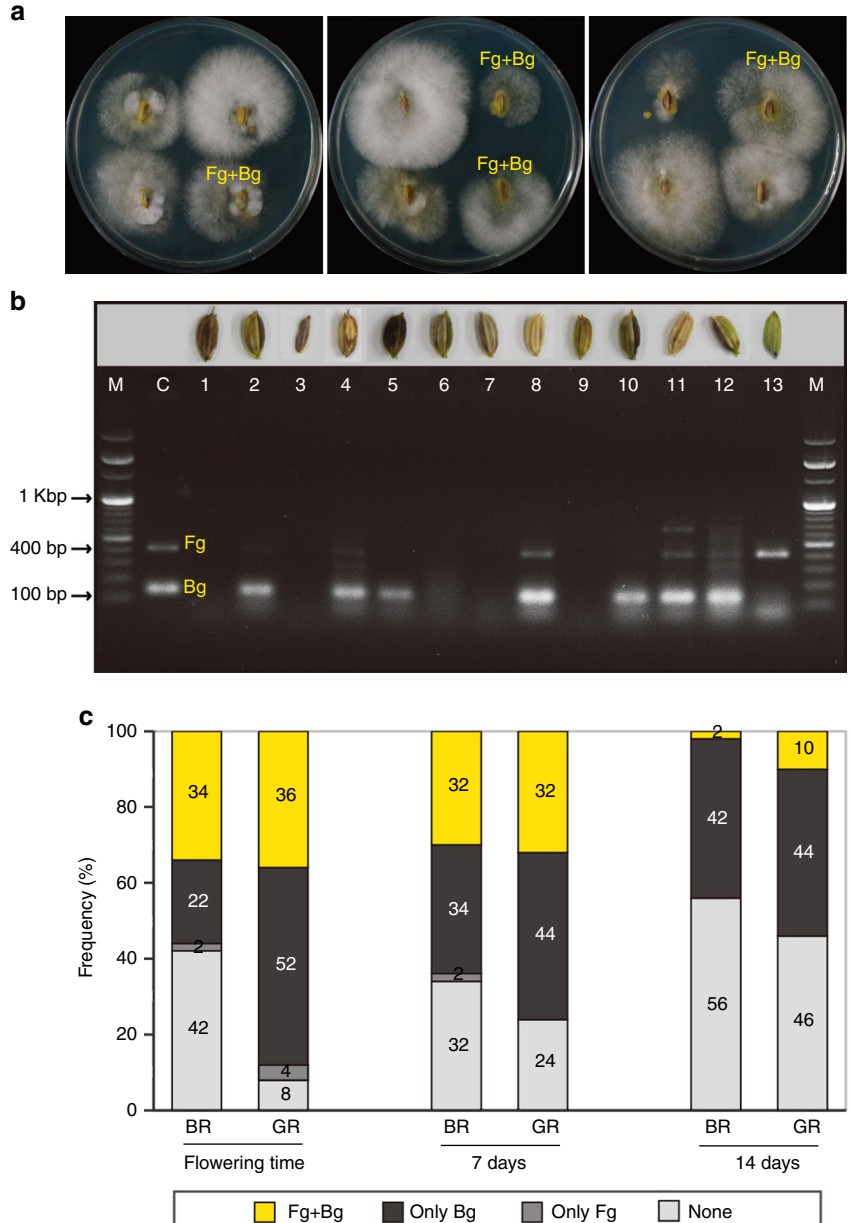

**Fig. 1** Co-detection of *Bukholderia glumae* (Bg) and *Fusarium graminearum* (Fg) in rice fields. **a** Both Bg and Fg were isolated from surface-sterilised rice grains that were placed on PDA for 3 d. **b** Bg and Fg were detected with specific primer pairs from diseased rice grains. Lane M, 100 bp marker; Lane C, positive control (Fg and Bg mixed DNA). **c** Coexistence frequencies of Bg and Fg in 50 rice grains collected at 7-d intervals beginning with the flowering season from field-grown black-type rice (BR) and glutinous-type (GR) rice were calculated. Bg and Fg were detected with specific primers

function of toxoflavin, a few lipid metabolism pathways were highly enriched. For example, fatty acid degradation was down-regulated in the WT strain compared to *ΔGzZC190*, yet was upregulated in the WT strain compared to *ΔGzC2H008*. Meanwhile, upregulation of fatty acid metabolism was observed in the WT strain compared to the *GzbZIP005* and *ΔGzC2H008* strains. In depth analyses of the WT strain compared to *ΔGzZC190* further showed that the greatest sensitivity to toxoflavin without any changes in tested phenotypes occurred in relation to an upregulation of genes involved in the fatty acid biosynthesis pathway (Supplementary Fig. 5).

**Lipid droplets are involved in toxoflavin resistance.** Based on the transcriptome analysis data described above, and previous

observations that lipid droplets (LDs) play an important role in tolerance to phytotoxins, fungicides and oxidative stress[21,22], we focused on fatty acid biosynthesis as a possible mechanism by which Fg develops resistance to toxoflavin. Therefore, lipid staining was initially performed of the WT and *ΔGzZC190* fungal cells. A dramatic increase in LD size was observed in the WT strain after toxoflavin treatment, while the LD size in the *ΔGzZC190* strain appeared unaffected by toxoflavin treatment (Fig. 2b). Previously, oleate was shown to increase LD formation in *Drosophila* S2 cells[23]. Therefore, we supplemented the MM containing toxoflavin with unsaturated fatty acids including oleic acid, linoleic acid, or linolenic acid and repeated the experiment. Large LDs were observed in all treated models, even in the hyphae of *ΔGzZC190*, and the vegetative growth of *ΔGzZC190* was restored to ~70% that of the WT strain (Fig. 2c, Supplementary

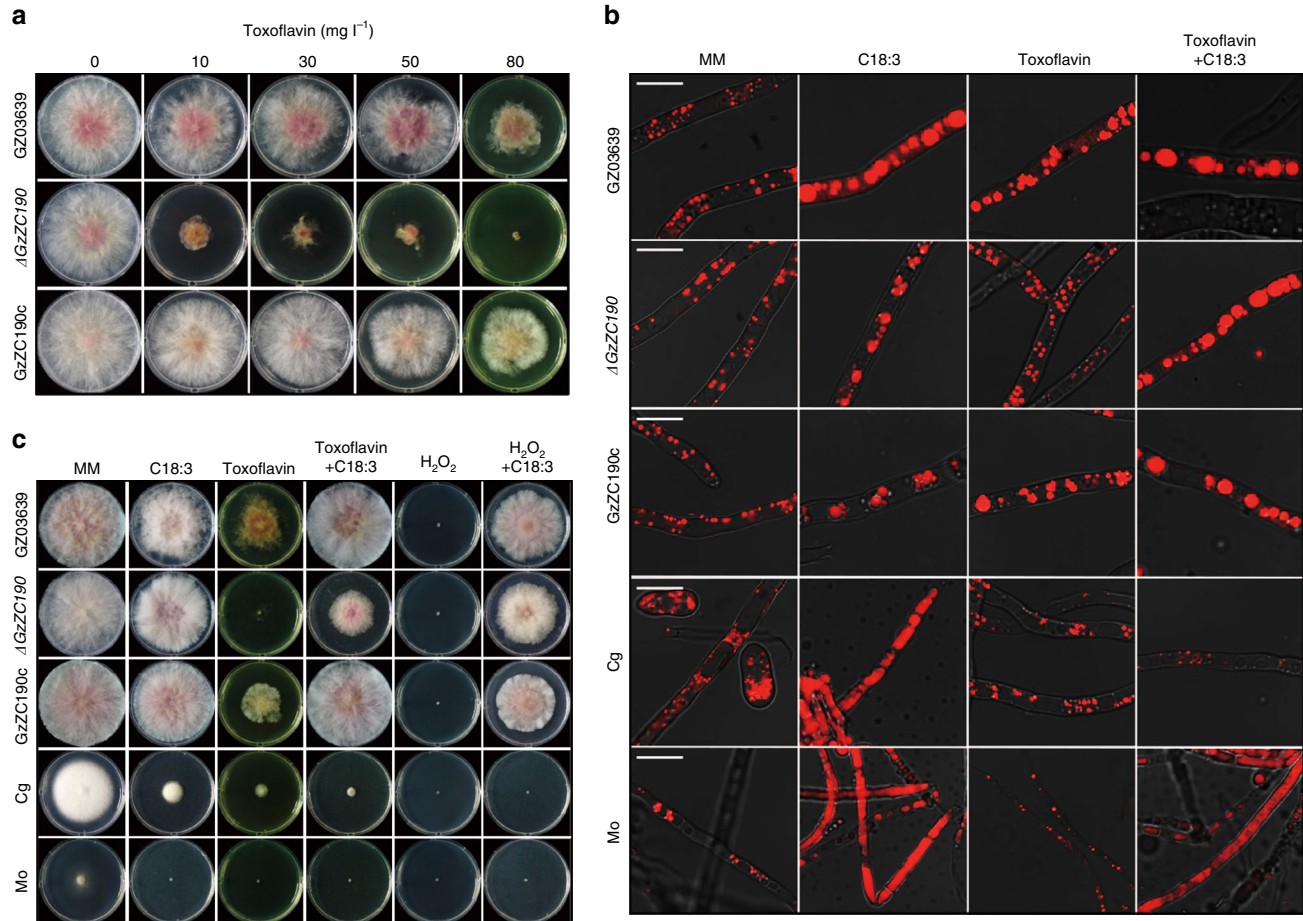

**Fig. 2** Resistance of Fg to toxoflavin. **a** Toxoflavin resistance was examined for Fg WT strain, GZ03639, for a toxoflavin-sensitive mutant derived from GZ03639 (*ΔGzZC190*), and for a *GzZC190*-complemented mutant (GzZC190c). Each strain was grown on MM supplemented with toxoflavin as indicated for 4 d. **b** Lipid staining was performed for GZ03639, *ΔGzZC190*, GzZC190c, *Colletotrichum gloeosporioides* (Cg) and *Magnaporthe oryzae* (Mo) cells. Mycelia were incubated for 24 h in MM containing toxoflavin and/or linolenic acid (C18:3) and stained with Nile Red. Scale bar, 10 μm. **c** Effect of linolenic acid, toxoflavin, and $H_2O_2$, in combination or individually, on the strains indicated that were grown for 4 d on supplemented MM

Fig. 6). As controls, LDs stained in *Colletotrichum gloeosporioides* (Cg) and *Magnaporthe oryzae* (Mo) hyphae were smaller than those of Fg (Fig. 2b). Moreover, when either oleic acid, linoleic acid or linolenic acid was added to Cg and Mo and the experiments were repeated, the toxoflavin sensitivity of Cg and Mo were unaffected (Fig. 2c, Supplementary Fig. 6). Furthermore, we also determined whether oleic acid, linoleic acid or linolenic acid enhance the resistance of WT to $H_2O_2$, since toxoflavin damages eukaryotic cells through the production of superoxide and $H_2O_2$[24,25]. Supplementation restored vegetative growth of the WT, *ΔGzZC190* and GzZC190c strains that was completely inhibited in the presence of 5 mM $H_2O_2$ (Fig. 2c, Supplementary Fig. 6).

**Triacylglycerides mediate toxoflavin resistance**. Due to the observed increase in LD size in the WT strain after toxoflavin treatment, triacylglyceride (TAG) content was measured. The goal was to obtain further insight into the relationship between increased LD size and toxoflavin-mediated ROS stress. LD has been shown to reduce cellular ROS stress via protection of unsaturated fatty acids[22,26,27]. Consequently, we compared the amounts of TAGs in the WT and *ΔGzZC190* strains with and without toxoflavin treatment. The total amounts of TAGs were higher in the *ΔGzZC190* strain than in the WT strain before toxoflavin treatment (ANOVA; $P < 0.05$). After toxoflavin

treatment, there were no statistically significant differences in the total TAGs between the WT and *ΔGzZC190* strains. A similar profile was observed for the amounts of TAGs containing oleic acid and the TAGs containing linoleic acid (Fig. 3a). Interestingly, the latter levels were reduced in the toxoflavin-treated *ΔGzZC190* strain compared with the toxoflavin-treated WT strain ($P < 0.05$) and the untreated *ΔGzZC190* ($P < 0.01$) strain (Fig. 3a, Supplementary Fig. 7). Correspondingly, genes involved in the biosynthesis of linolenic acid were found to be upregulated in the WT strain compared to the *ΔGzZC190* strain after toxoflavin treatment, whereas most genes were downregulated in the toxoflavin-treated *ΔGzZC190* strain compared to the untreated *ΔGzZC190* strain (Fig. 3b). These results are consistent with the observed changes in the TAGs containing linolenic acid in response to toxoflavin.

**Superoxide dismutase 1 mediates toxoflavin resistance**. The vegetative growth of mutant strains carrying deletions of *FgAP*, *FgSKN7*, *FgATF1*, or *FgELP1* has been associated with oxidative stress[28–31]. Here, the growth of these mutant strains was not suppressed on toxoflavin-supplemented media (Supplementary Fig. 8a). However, vegetative growth of the superoxide dismutase 1 mutant (*Δsod1*)[32] was inhibited similar to that of the *ΔGzZC190* strain (Supplementary Fig. 8a). Interestingly, catalase supplementation of the *ΔGzZC190* strain did not restore vegetative

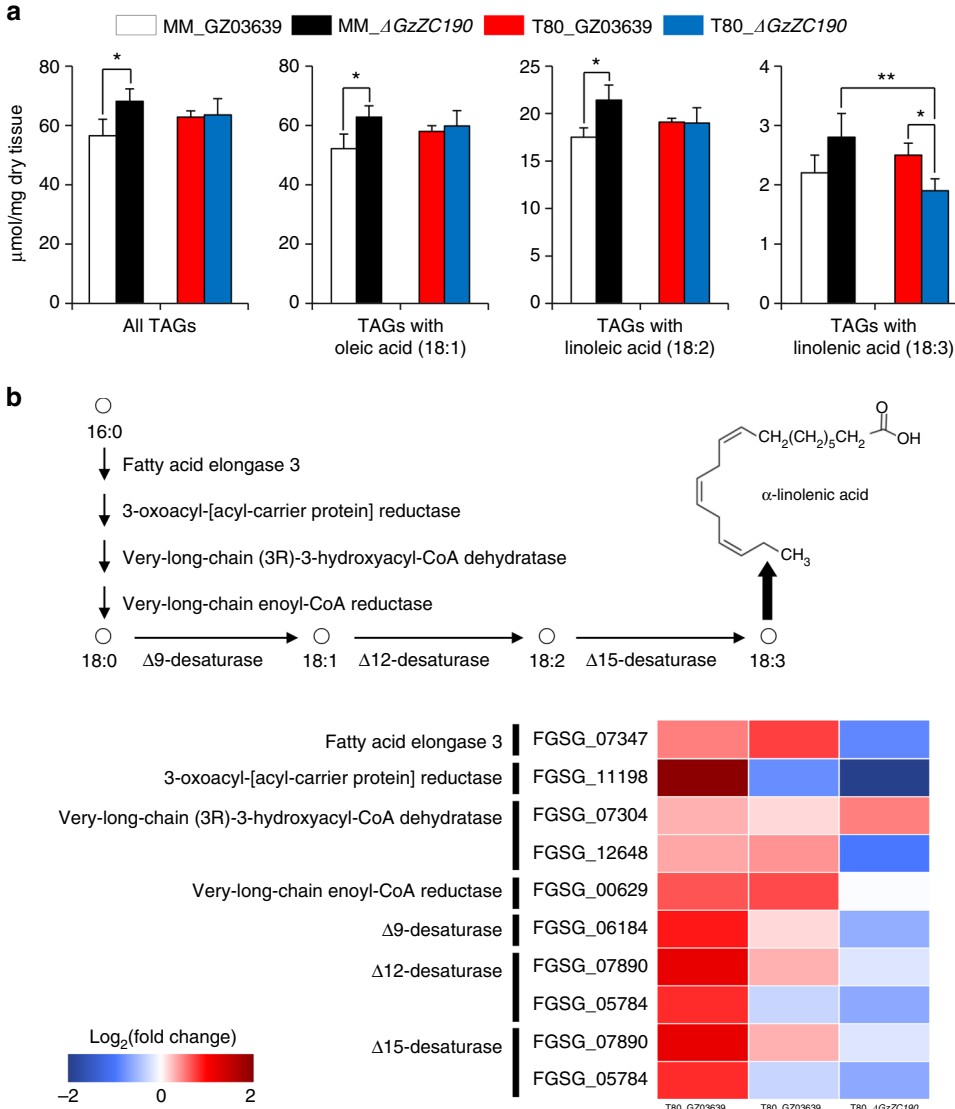

**Fig. 3** Quantification of mycelia TAGs by LC–MS/MS and expression patterns of genes involved in the biosynthesis of linolenic acid (18:3). **a** Dry mycelia of GZ03639 and $\Delta$GzZC190 ($n = 3$ each) were incubated for 24 h in MM with and without toxoflavin before TAGs were detected with LC–MS/MS. The data presented are the mean ± s.d. (ANOVA; *$P < 0.05$, **$P < 0.01$). **b** RNA-seq analyses were performed to obtain expression profiles of genes involved in the production of linolenic acid (18:3) from palmitic acid (16:0). The KEGG database (http://www.kegg.jp/) and manual annotation based on orthologs of other species were used to annotate genes involved in the production of linolenic acid

growth that was inhibited by toxoflavin, yet it did restore vegetative growth that was inhibited by $H_2O_2$ (Supplementary Fig. 8b).

In $\Delta$GzZC190, transcript levels of all five *sod* genes were upregulated. However, following toxoflavin treatment, all five genes were downregulated. The opposite profile was obtained in the WT strain, with the transcripts of all five *sod* genes being upregulated after toxoflavin treatment (Supplementary Fig. 9a). Transcript levels of genes encoding catalases were also upregulated in the WT strain after toxoflavin treatment, yet a distinct pattern was not observed in the $\Delta$GzZC190 strain (Supplementary Fig. 9b).

**Bg enhances Fg spore production**. To address the ecological significance of the interaction between Fg and Bg, we conducted several experiments to determine whether chemical or physical interactions exist between these two pathogens. Compared to the cultivation of Fg alone, co-cultivation resulted in ~10-fold greater

fungal spore production. Addition of a cell-free supernatant of a Bg (BGR1) culture to Fg (GZ03639) further increased spore production (Fig. 4a). In contrast, filtrates of *Pseudomonas syringae* (DC3000) and *Bukholderia pyrrocinia* (CH67) cultures did not enhance spore production (Fig. 4a). When toxoflavin alone or $H_2O_2$ were added to GZ03639, spore production also increased, albeit less than that of the bacterial culture filtrate (Fig. 4b, Supplementary Fig. 9a). There were three Fg field strains which also exhibited higher spore production with toxoflavin (Supplementary Fig. 10b). Gene expression profiling of the WT strain with toxoflavin correspondingly resulted in higher expression of 24 genes related to sporulation (Supplementary Table 5).

**Toxoflavin increases production of deoxynivalenol by Fg**. A plant's response to fungal infection includes an oxidative burst, and this in turn can induce the production of mycotoxins in fungi[33]. For example, ROS have been shown to trigger aflatoxin production in *Aspergillus parasiticus*[34], and in Fg, production of a

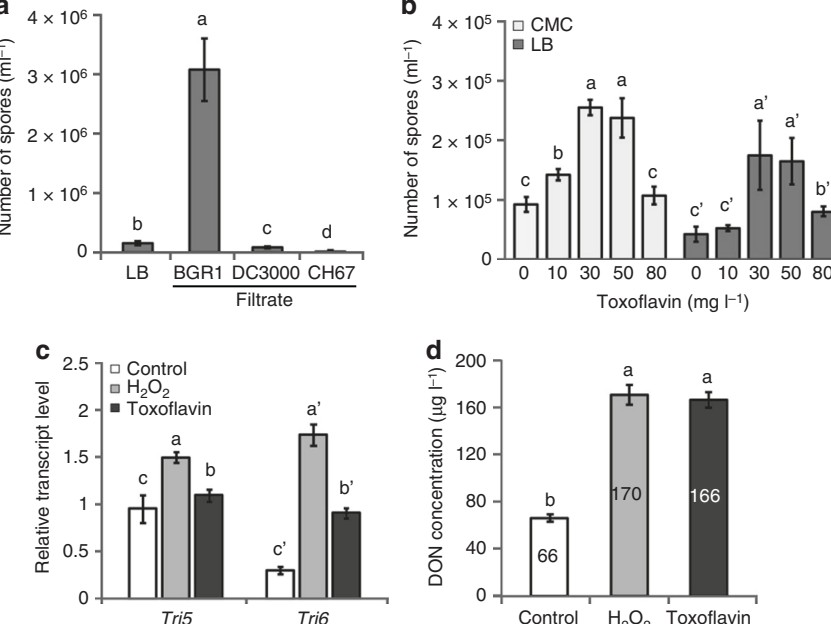

**Fig. 4** Effects of toxoflavin on the generation of spores and DON production. **a** Effect of bacterial filtrates on Fg spore production. WT strain GZ03639 was incubated with bacterial culture filtrates for 24 h. BGR1: Bg, DC3000: *Pseudomonas syringae*, CH67: *Burkholderia pyrrocinia*. **b** Effect of toxoflavin on Fg spore production. GZ03639 was incubated for 24 h in CMC or LB medium supplemented with toxoflavin at the concentrations indicated. **c** Effect of toxoflavin on transcription of the trichothecene biosynthesis genes, *Tri5* and *Tri6*, in GZ03639 grown in GYEP medium (control) and in the presence of toxoflavin (80 mg l$^{-1}$) or $H_2O_2$ (1 mM). **d** Effect of toxoflavin on Fg DON production in GYEP medium (control) and in the presence of toxoflavin (80 mg l$^{-1}$) or $H_2O_2$ (1 mM). Bars not sharing a letter are significantly different according to Tukey's test ($P < 0.05$, $n = 3$) and data presented are the mean ± s.d

type B trichothecene, deoxynivalenol (DON), is triggered[35,36]. Correspondingly, the genes, *Tri5* and *Tri6*, which encode proteins with functions important to DON production in Fg, were found to be upregulated in the presence of 1 mM $H_2O_2$ or 80 mg l$^{-1}$ toxoflavin in our studies (Fig. 4c). These conditions also resulted in greater production of DON (Fig. 4d).

**Co-cultivation results in a physical attachment between Bg and Fg.** To understand the benefit, if any, of greater fungal spore production to the bacterium, Bg, physical interactions between the pathogens were examined. In a nested plate system designed to monitor physical attachment[37], both BGR1 and DC3000 grew well on a central plate embedded in media inoculated with Fg. However, only BGR1 was found outside of the central plate, and it was physically attached to fungal hyphae and spores (Fig. 5a, b). While the signalling molecules involved remain to be isolated, these results demonstrate that BGR1 is able to move towards GZ03639. A similar observation was made when a greater number of BGR1 cells were found to have moved into a capillary tube containing a culture filtrate of GZ03639, compared with other capillaries that contained pure medium or a culture filtrate of another rice fungal pathogen, *M. oryzae* (Fig. 5c). Furthermore, the Fg culture filtrates did not attract the DC3000 cells (Fig. 5c).

**Co-inoculation increases disease severity and DON production.** A single inoculation of BGR1 or GZ03639 on rice heads in the late-flowering stage resulted in disease severities of 15 and 33%, respectively. In contrast, when both pathogens were co-inoculated, the disease severity was 82% and disease progression to the stem was observed (Fig. 6a, Supplementary Fig. 11). In addition, both Bg and Fg were detected in symptomatic grains of 34 out of 36 (94%) untreated rice plants that were adjacent to the co-inoculated rice plants. DON production was also ~twofold higher in the rice plants that were inoculated with both Fg and Bg

compared with the rice plants that received a single-Fg inoculation (Fig. 6b). The Fg-only rice plants also had many hyphae, while spores were rarely observed. In contrast, both hyphae and spores were observed on the rice plants inoculated with both Fg and Bg, with Bg cells frequently present on the hyphae and spores (Fig. 6c).

**Physical attachment of Bg to Fg protects Bg from UV stress.** In addition to dispersion benefits, the Fg spores appear to provide protection for Bg from ultraviolet (UV) light-induced damage. When BGR1 cells and GZ03639 spores were exposed to UV germicidal lamps for 1 and 5 min, respectively, both pathogens were completely killed. In contrast, when BGR1 cells were co-cultivated with GZ03639 spores and then were exposed to UV light for 5 min, colony-forming BGR1 was recovered, although the GZ03639 spores were completely unviable (Supplementary Fig. 12, Supplementary Table 6).

**Discussion**
Our efforts to identify a mechanism by which Fg is able to tolerate the production of toxoflavin by Bg on rice heads started with the characterisation of three Fg mutants carrying deleted TFs that exhibited sensitivity to toxoflavin, *ΔGzZC190, ΔGzbZIP005* and *ΔGzC2H008*. To identify possible links among their genes in relation to toxoflavin resistance, transcriptome data were obtained for each, and for the parental wild-type strain, in the presence and absence of toxoflavin. Interestingly, many of the DEGs that were identified were enriched in pathways involving fatty acid degradation, fatty acid biosynthesis and fatty acid metabolism. Toxoflavin treatment was also found to induce the production of larger LDs in Fg cells. LDs have previously been shown to reduce cellular ROS stress via protection of unsaturated fatty acids[22,26,27]. In addition, LDs serve as the principal reservoirs for storing cellular energy and contribute to the repair of

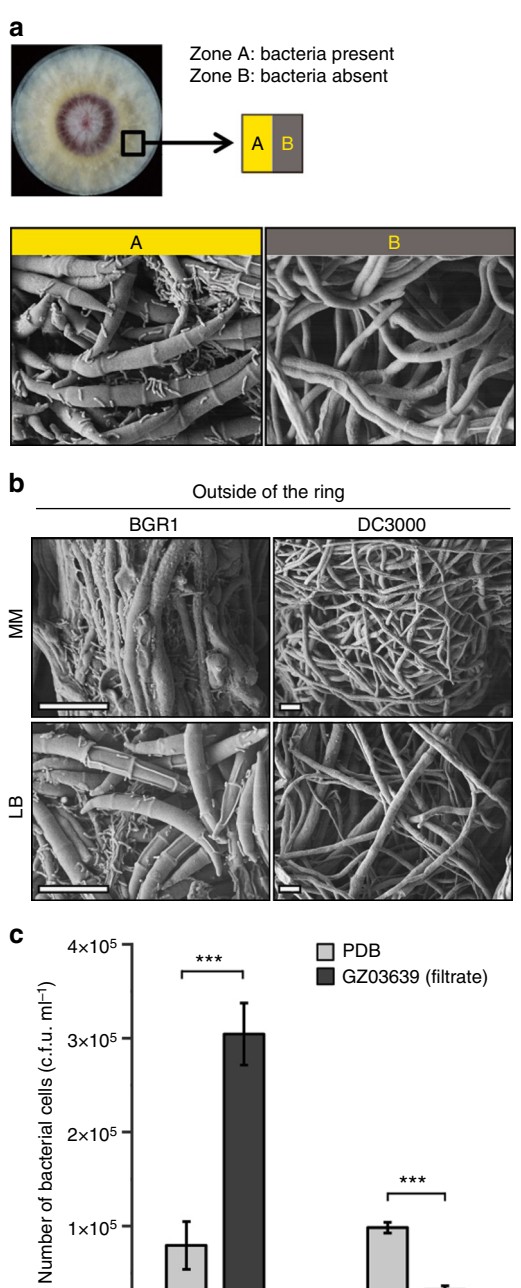

**Fig. 5** Physical association between Bg and Fg. **a** Bacteria and fungi were co-inoculated inside a culture plate (35 × 10 mm) containing LB agar for up to 10 d in the dark at 30 °C. A field emission scanning electron microscope detected spores in the zone where bacterial cells were present (A) and not in the zone where bacterial cells were absent (B). Scale bar, 10 μm. **b** Physical association between Fg and Bg. The negative control DC3000 did not show transported hyphae outside of the ring. **c** Capillary tubes filled with PDB or GZ03639 filtrate were placed into the wells of 96-well plates containing BGR1 or DC3000 bacterial cells. After incubating the plates at 30 °C for 3 h, liquid in the capillaries were spread onto LB agar plates and colonies formed after 24 h were counted. The experiment was repeated twice with three replicates per sample and data presented are the mean ± s.d. (ANOVA; ***$P < 0.001$)

destroyed membrane lipids in eukaryotic cells[23], during *Drosophila* development LDs limit the levels of ROS[22], and in fungi LDs trap phototoxins to serve as a resistance mechanism[21]. To further investigate the molecular details of the mechanism mediating toxoflavin resistance in Fg, the TAG content of the WT and Δ*GzZC190* strains with and without toxoflavin treatment was investigated with LC–MS/MS. Toxoflavin treatment of Δ*GzZC190* was found to reduce the levels of TAGs containing linolenic acid, and these results were consistent with the observation that genes associated with fatty acids containing α-linolenic acid were upregulated in response to toxoflavin (Fig. 3). Taken together, these results suggest that TAGs containing linolenic acid are protected within LDs, and the genes involved in the biosynthesis of linolenic acid are induced in response to ROS generated from toxoflavin. Further support for this model was provided by the treatment of Δ*GzZC190* with linolenic acid (or linoleic acid) and the subsequent restoration of toxoflavin sensitivity. Thus, TAGs containing linolenic acid may play a role in resistance to ROS stress that is generated in response to toxoflavin. Furthermore, the previous observations that vegetative growth and sporulation in *Aspergillus flavus* were enhanced after treatment with auto-xidated linolenic acid[38] suggests that linolenic acid may have additional roles in Fg condia formation after toxoflavin treatment.

When vegetative growth of the superoxide dismutase 1 mutant (Δ*sod1*)[32] was found to be similar to that of the Δ*GzZC190* strain, all five *sod* genes of Fg were investigated and all of them were found to be downregulated following toxoflavin treatment (Supplementary Fig. 9). Further examination of these results in the present study suggests that superoxides that are produced following toxoflavin treatment may be converted into $H_2O_2$, which could be subsequently converted into water by catalases in the WT Fg strain. These observations, in combination with the previous observation that chlamydospore-like structures form and accumulate lipids that are associated with resistance to oxidative stress in Fg[39], suggest that LDs containing linolenic acids have the potential to rapidly scavenge superoxides that are generated by toxoflavin to provide toxoflavin resistance.

ROS have been shown to play an important role in the cell differentiation of fungi and other eukaryotes[40,41]. In addition, transient increases in ROS levels have been shown to induce a transition between vegetative growth and asexual reproduction[42]. A model in which increased spore production by Fg is related to ROS production triggered by toxoflavin is supported by the present results, since both toxoflavin and $H_2O_2$ were shown to cause a statistically significant increase in Fg spore production (Supplementary Fig. 10), and 24 sporulation-related genes were highly upregulated by toxoflavin (Supplementary Table 5), including *FGSG_01877* and *FGSG_01915* which were identified as regulators for sporulation in Fg. Accordingly, when these two genes were deleted, spore production was decreased[43,44]. Conversely, when *FGSG_01877* was overexpressed, a substantial increase in spore production was observed in Fg[43]. Thus, the present results indicate that toxoflavin enhances the expression of two regulators of sporulation (*FGSG_01877* and *FGSG_01915*) and this increases spore production in Fg.

Bg can be present prior to the arrival of Fg spores on rice because Bg is seed-borne and can move to the rice panicle as the plant grows. We propose that the presence of Bg on rice is beneficial to Fg for the following reasons. First, Bg enhances production of DON by Fg, which aids in disease progression. Previously, ROS were found to enhance DON production in Fg[35,36], and this observation is consistent with the observation that DON production increased following toxoflavin treatment. While DON production is not essential for the initial infection of Fg in wheat plants (yet it helps systemic colonisation in wheat spikelets[45]), co-inoculation of Fg and Bg resulted in greater

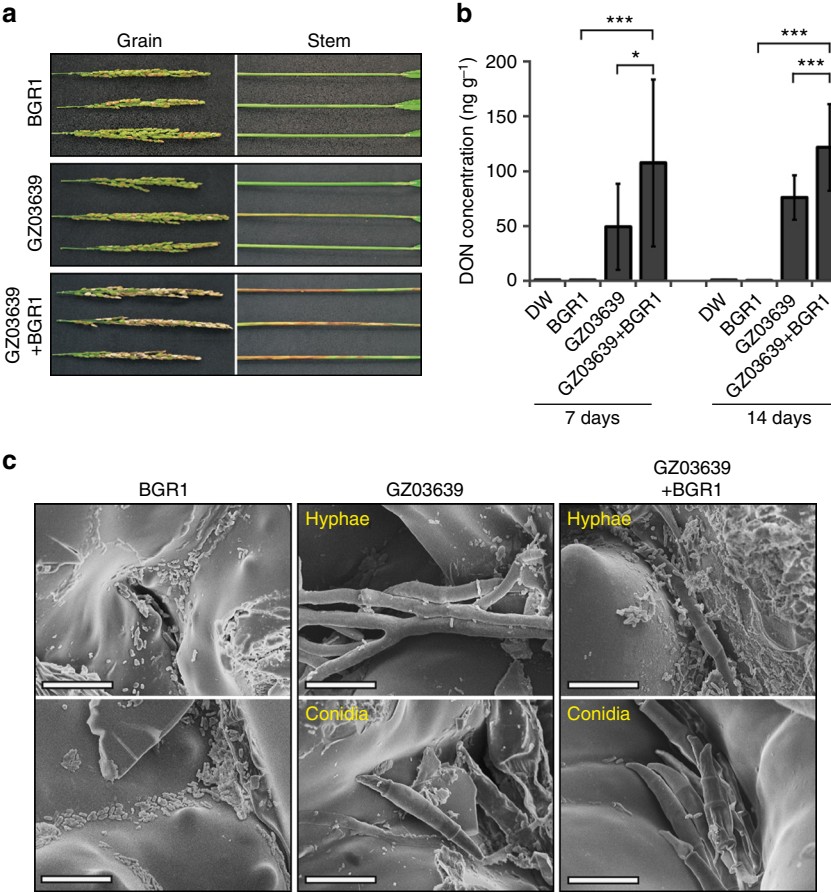

**Fig. 6** Interactions between Bg and Fg on rice plants. **a** Disease severity of rice plants inoculated with either Bg, Fg or a combination of both. BGR1 and GZ03639 indicate single inoculations with Bg and Fg, respectively. GZ03639 + BGR1 indicates a co-inoculation of both pathogens. **b** DON was quantified in rice heads inoculated with either Bg, Fg, or a combination of both 7 d and 14 d after inoculation. The experiment was repeated twice with three replicates per sample and data presented are the mean ± s.d. (ANOVA; *$P < 0.05$, ***$P < 0.001$). **c** Images of Fg and Bg in rice husks as observed with a field emission scanning electron microscope. Scale bar, 10 μm

disease severity and higher DON production in infected rice heads (Fig. 6, Supplementary Fig. 11). Second, Bg can prevent the growth of other fungal competitors of Fg because other saprophytic and pathogenic fungi are sensitive to toxoflavin. Fungi must compete with various microorganisms to occupy an area, and on a plant surface there can be a diverse set of microorganisms. Toxoflavin exhibits a broad range of antibiotic properties that can affect eukaryotes[46]. Therefore, the presence of Bg can facilitate the occupation of rice heads by Fg since other microorganisms are sensitive to the production of toxoflavin.

An interesting observation from the present study was that the physical attachment of Bg cells to Fg spores protected the Bg cells from UV light-induced damage. Specifically, the present results demonstrate that the attachment of bacteria to conidia resulted in a marked increase in resistance to UV stress compared to unattached bacteria. Physical attachment to Fg also facilitated aerial movement of the bacterium for its dispersal. Generally, the migration of bacteria is very limited and often requires water for dispersal. In contrast, the attachment of bacteria to fungal hyphae provides a more rapid and expansive route of dispersal. Correspondingly, recent reports have indicated that fungi are essential for the colonisation and dispersal of *Burkholderia* species[9,37,47]. Thus, cooperative interactions between Bg and Fg result in the dispersal of Bg, as well as protection of Bg from UV stress.

The identification of this BFI that accounts for the observed exacerbation of disease frequency and severity in rice fields provides an opportunity to target this association and attenuate its

virulence. For example, ROS scavenging or lipid depletion could potentially interfere with this cooperative relationship to reduce disease severity. Moreover, further examination of the evolutionary history of the genes involved may lead to the identification of other systems that may have evolved cooperative interactions.

## Methods

**Strains and culture conditions.** *Fusarium graminearum* (Fg) wild-type (WT) strain, GZ03639, and transcription factor deletion mutants (TFs)[19] derived from this WT strain were used in this study. TF complementation construct was generated by the double-joint (DJ) PCR method[48]. The primers used for genetic manipulation were listed in the Supplementary Table 7. *Burkholderia pyrrocinia* (CH67) and *Pseudomonas syringae* (DC3000) were used as negative controls. Fg field strains, SCK04, GWR12R5 and GWS16-4-7, were obtained from Soonchunhyang University (Cheonan, Korea), while other fungal and bacterial species were obtained from the Centre for Fungal Genetic Resources (Seoul National University, Seoul, Korea). The bacterial strains were cultured in lysogeny broth (LB) medium at 30 °C with shaking at 200 r.p.m. for 24 h; the fungal strains were maintained on potato dextrose agar (PDA) at 25 °C; and spores were induced in carboxymethyl cellulose (CMC)[49]. Spore germination was tested in minimal medium (MM). All strains used in this study were stored in 15% (v/v) glycerol at −70 °C.

**Detection of Bg and Fg from diseased rice grains.** Rice grains with head blight symptoms were collected from a southern province in Korea (Gyeongnam) during August 2015 at intervals of 7 d beginning with the flowering times of black- and glutinous-type rice. Genomic DNA was extracted from 50 rice grains collected from each cultivar, and the presence of Bg or Fg was tested by conventional PCR with *F. graminearum*-specific primers (Fg16F; 5′-CTCCGGATATGTTGCGTC AA-3′, Fg16R; 5′-GGTAGGGTATCCGACATGGCAA-3′)[50] and a *B. glumae* primer set designed for a *rhs* family gene (BG1F; 5′-CCGCGGCTGTTCATGAGGGATAA- 3′, BG1R; 5′-CGGGCGGAACGACGGTAAGT-3′)[51]. The

predicted PCR product lengths were 420 and 138 bp, respectively. The PCR products were purified from gels for sequencing (Macrogen, Seoul, Korea).

**Isolation of a toxoflavin-sensitive Fg mutant**. Toxoflavin was added into autoclaved MM cooled to 50 °C at concentrations of 0, 10, 30, 50 and 80 mg l$^{-1}$. To identify toxoflavin-sensitive mutants, all 657 TFs[19] were incubated on MM at 25 °C for 3 d. Mutant strains exhibiting slow vegetative growth were selected. To test the effects of toxoflavin on spore germination, spores of Fg strains were incubated in MM liquid containing 0 or 80 mg l$^{-1}$ toxoflavin, and germination was observed 0, 4 and 8 h later[18].

**RNA preparation and RNA-Seq analysis**. GZ03639 and three toxoflavin-sensitive mutant strains were grown in 50 ml of complete medium (CM) at 25 °C with shaking at 200 r.p.m. for 72 h. Filtered mycelia were incubated in 50 ml of MM containing toxoflavin (80 mg l$^{-1}$) for 24 h at 200 r.p.m. (9 g) for RNA preparation. The fungal mycelia were collected and washed twice with distilled water before being ground in liquid nitrogen. Total RNA was extracted with a NucleoSpin RNA Plant Kit (MACHEREY-NAGEL GmbH & Co. KG, Düren, Germany), according to the manufacturer's protocol. Sequencing was performed with an Illumina HiSeq2000 instrument (Illumina, San Diego, CA, USA) at the National Instrumentation Centre for Environmental Management (NICEM, Seoul, Korea). The raw data discussed in this study have been deposited in Gene Expression Omnibus of NCBI (accession number: GSE104623).

Relative transcript abundance was measured in reads per kilobase of exons per million mapped reads[52]. Differentially expressed genes (DEGs) were identified using the DEGseq software package in the R statistical environment[53]. In this analysis, the MA plot-based method was employed, and the filtering threshold included $P < 0.05$ and an absolute value ≥twofold change. All DEGs were annotated according to Gene Ontology (GO; http://geneontology.org/) and Kyoto Encyclopedia of Genes and Genomes (KEGG; http://www.genome.jp/kegg/) databases to establish functional categories. Enrichment of the functional categories was performed by using a hypergeometric distribution[54]. $P$-values <0.01 and <0.05 were considered to indicate significance in the GO and KEGG analyses, respectively. The network system of the fatty acid biosynthesis pathway was constructed with Cytoscape software (www.cytoscape.org/).

**Lipid body staining and oleate test**. Mycelia were incubated in MM containing 80 mg l$^{-1}$ toxoflavin and 1% sodium oleate, 1% linoleic acid, or 1% linolenic (Sigma-Aldrich) at 25 °C with shaking at 200 r.p.m. for 24 h. Lipid droplets in fungal cells were stained with a Nile Red solution consisting of 50 mM Tris/maleate buffer (pH 7.5), 20 mg ml$^{-1}$ polyvinylpyrrolidone, and 2.5 μg ml$^{-1}$ Nile Red Oxazone (Sigma-Aldrich). The samples were incubated for 15 min at room temperature and were washed two times with phosphate-buffered saline. Fluorescence emitted by the lipid droplets was observed with a Carl Zeiss confocal microscope (excitation at 490 nm, LSM 510). To determine whether vegetative growth that was affected by toxoflavin could be complemented with exogenous sodium oleate, 1% sodium oleate was added into MM agar supplemented with 80 mg l$^{-1}$ toxoflavin or 5 mM $H_2O_2$. Each fungal strain was incubated at 25 °C for 4 d.

**Lipid extraction**. Mycelia of GZ03639 and Δ*GzZC190* were incubated in 10 ml of MM containing toxoflavin (80 mg l$^{-1}$) for 24 h at 200 r.p.m. The fungal mycelia were subsequently washed twice with distilled water and lyophilised for 2 d. Lipids were extracted from the freeze-dried mycelia according to a modified Matyash method[55,56]. Briefly, 400 μl of methanol:water (3:1) containing 0.1% butylated hydroxytoluene was added to the samples. After a homogenisation step with Tissue Lyser (30 Hz (1/s), 30 s; QIAGEN, Waltham, MA, USA), methyl-tert-butyl ether (1 ml) was added and the samples were vortexed for 1 h. Solvent partitioning was achieved with the addition of 250 μl of water, and the upper phase was transferred to a new tube, dried under vacuum, and reconstituted with 100 μl chloroform: methanol (1:9, v/v) containing 20 ng ml$^{-1}$ internal standard triacylglycerides (TAG) (15:0/15:0/15:0).

**Triacylglyceride analysis**. Liquid chromatography–tandem mass spectrometry (LC–MS/MS) analyses were performed with a triple quadrupole mass spectrometer (LC–MS 8040; Shimadzu, Kyoto, Japan) coupled to a Nexera2 LC system (Shimadzu) using a Kinetex C18 column (100 × 2.1 mm, 2.6 μm; Phenomenex, Torrance, CA, USA). Mobile phase A was 10 mM ammonium acetate in water: methanol (1:9, v/v) and mobile phase B was 10 mM ammonium acetate in methanol:isopropanol (1:1, v/v). To achieve the best separation of TAG, a gradient elution was conducted as follows: 30% B (0 min), 95% B (15–20 min), and 30% B (20–25 min). The flow rate was 200 μl min$^{-1}$. Quantitation was performed with selected reaction monitoring of the $(M + NH_4^+)$ ion and related product ions for each TAG[57,58].

**Quantitative real-time PCR (qRT-PCR)**. To compare the transcript levels of genes, WT and Δ*GzZC190* were grown in 50 ml of complete medium (CM) at 25 °C with shaking at 200 r.p.m. for 72 h. The filtered mycelia were incubated in 50 ml of MM containing toxoflavin (80 mg l$^{-1}$) for 24 h at 200 r.p.m. and total RNA was

extracted using NucleoSpin RNA Plant Kit (MACHEREY-NAGEL GmbH & Co. KG, Düren, Germany). The amounts of transcripts were quantified by qRT-PCR with specific primers (Supplementary Fig. 7). Quantification values were automatically determined using Bio-Rad CFX Manager, version 3.1, and the threshold cycle (Ct) values were determined. The experiment was repeated twice, with three replicates for each repeat, and final Ct values are presented as an average[59]. To determine whether vegetative growth affected by toxoflavin can be complemented with exogenous catalase (Sigma-Aldrich), 1000 unit was added into MM agar supplemented with 80 mg l$^{-1}$ toxoflavin or 5 mM $H_2O_2$, and each fungal strain was incubated at 25 °C for 4 d.

**Induction of Fg spore production**. Asexual spores ($10^5$ ml$^{-1}$) of Fg were inoculated in 50 ml of potato dextrose broth (PDB) and incubated at 25 °C on a rotary shaker at 200 r.p.m. After 24 h, mycelia were collected by filtration through Miracloth (Calbiochem) and were washed twice with sterile distilled water. Freshly collected mycelia were then inoculated in 20 ml CMC containing 10, 30, 50 or 80 mg l$^{-1}$ toxoflavin and incubated at 25 °C with shaking at 200 r.p.m. After 24 h, the numbers of spores produced were counted with a haemocytometer.

**Transcription of *Tri* genes and trichothecene production**. To test the effect of toxoflavin on trichothecene biosynthesis, mRNA levels of the trichothecene biosynthesis genes, *Tri5* (FSGG_03537, trichodiene synthase) and *Tri6* (FSGG_03536, transcription factor for trichothecene synthesis), as well as trichothecene production, were analysed. Briefly, each strain was incubated in 50 ml GYEP (10 g of glucose, 1 g of yeast extract, and 1 g of peptone in 1 l)[35] supplemented with or without toxoflavin (80 mg l$^{-1}$) at 25 °C on a rotary shaker (200 r.p.m.) for 5 d. Total RNA was purified from the mycelia using a NucleoSpin RNA Plant Kit (MACHEREY-NAGEL GmbH & Co. KG), according to the manufacturer's protocol. The amounts of *Tri5* and *Tri6* transcripts were quantified by qRT-PCR with specific primers for each gene[60]. Concentrations of DON in the culture filtrates were quantified with an enzyme-linked immunosorbent assay kit (CUSABIO, College Park, MD, USA), according to the manufacturer's instructions[61].

**Attachment assay of bacteria on fungal hyphae and spores**. Dispersal and attachment of bacteria on fungal hyphae and spores was monitored[37] with a slight modification. Briefly, sterile tissue culture plates (35 × 10 mm) were used as a physical barrier for bacterial cells during the pouring of LB or MM agar into Petri dishes (90 × 15 mm) up to 5 mm below the edge of the smaller plate. Selected fungi and *Burkholderia* strains were inoculated onto the inner side of the smaller plate, so that only bacterial cells which could attach and/or disperse would be detected on the outside of the smaller plate. Different regions of the cultures were observed under a field emission scanning electron microscope (JSM-6700F) after preparation[62].

**Assay of Bg attraction to Fg**. An attraction assay was performed according to the method used for *P. syringae*[63], with a slight modification. Briefly, bacteria cells ($OD_{600}$ of 0.1) were grown in LB at 30 °C and then were diluted $10^2$-, $10^3$- and $10^4$-fold. An aliquot (100 μl) of each diluent was then added to the wells of a 96-well plate. Fungal spores ($10^5$ ml$^{-1}$) were incubated in PDB for 3 d, and the culture was filtered through a 0.2-μm syringe filter. Capillary tubes filled with the filtrate were placed into the 96-well plate containing serial dilutions of the bacterial cells, and the plate was subsequently incubated at 30 °C for 3 h. Fungal filtrates in the capillary tubes were allowed to spread on LB agar plates at 30 °C. After 24 h, the numbers of colonies that grew were counted and recorded. The experiment was repeated twice with three replicates per sample.

**UV light stress test**. Bacterial cells, fungal spores and fungal spores with bacterial cells were exposed to a germicidal UV light (40 W; G40T10) for 0, 1, 3, and 5 min, respectively. To induce attachment of the bacterial cells to the spores, Fg and Bg were co-incubated on LB agar at 30 °C for 10 d. The spores were then collected by filtration through Miracloth and diluted with distilled water to $10^2$ ml$^{-1}$. Each diluent (100 μl) was subsequently spread on an LB agar plate and incubated at 30 °C. After 24 h, the plates were exposed to UV light as indicated above and the numbers of bacterial colonies present were counted and recorded.

**Disease severity following co-infection of Bg and Fg**. Briefly, rice (*Oryza sativa* L. cv. Dongjin) heads were dipped into suspensions of Bg ($10^7$ cells per ml) or Fg ($10^6$ spores per ml) for 1 min and then were individually sealed in plastic bags for 72 h. The infected plants were then placed in a greenhouse and rice grains exhibiting blight symptoms were counted 14 d after inoculation. Disease severity was calculated based on the number of diseased grains per rice head. For the DON analysis, 40 spikes were dried, pulverised, and then analysed by an enzyme-linked immunosorbent assay method.

**Statistical analysis**. The post hoc Tukey test (implemented in R version 3.1.2 software) was used to test the significance of differences among the mean values for lipid content, bacterial colony forming units, spore production, DON production and disease severity between treatments.

**Data availability**. The RNA-seq data have been deposited in the NCBI Gene Expression Omnibus database with accession code GSE104623. Other relevant data supporting the findings of the study are available in this published article and its Supplementary Information files, or from the corresponding authors upon request.

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

## Acknowledgements

We thank Prof. Yin-Won Lee (Seoul National University, Korea) for providing transcription factor deletion library and for discussions. This work was supported by grants from the Strategic Initiative for Microbiomes in Agriculture and Food, Ministry of Agriculture, Food and Rural Affairs, Republic of Korea (No. 916009021SB010), and from the Next-Generation Bio Green21 Program, the Rural Development Administration, Republic of Korea (No. PJ0111802016).

## Author contributions

Y.S. and J.L. conceived the study and designed the experiments. B.J., J.P., N.K., T.L., So. K., J.K., I.K., Y.K., K.-H.Y., Y.C., S.J., K.S.L., B.Y.K., J.C.S., W.C.K. and D.Y. performed experiments. B.J., J.P., H.-H.L., L.B., K.-H.L., Su.K., Y.-S.S. and J.L. analysed the data. B.J., L.E.B., Y.-S.S. and J.L. wrote the paper. All authors provided comments on the paper.

## Additional information

**Competing interests:** The authors declare no competing financial interests.

