## [Peer Review File · Nature Communications]

Reviewer #1 (Remarks to the Author):

Manuscript entitled 'Cooperative interactions between seed-borne bacterial and air-borne fungal pathogens on rice'

This manuscript presents studies on the presence and role of a fungal (*Fusarium graminearum*) and bacterial pathogen (*Burkholderia glumae*) on a rice grain disease. The two pathogens were found to co-exist in the wild in rice grains and in causing a more severe/aggressive disease when co-inoculated. Other studies indicate that the two microorganisms can interact, stimulate production of toxins and the bacterium can live on the fungal spores. A major conclusion is that the fungus and bacterium can co-exist and cooperate.

General comment

This study presents some interesting data on the interaction between *Fusarium graminearum* and *Burkholderia glumae*. Some of the data is solid whereas some is less and is at its early stages being rather preliminary. Studies deciphering interactions between fungi and bacteria are at large unknown, therefore this is a relatively novel field. Overall this study has some interesting and novel results but it lacks focus presenting in several cases preliminary data from several angles. See specific comments below.

1. The screening of *Fusarium* mutants resistant to toxoflavin yielded one mutant, a putative regulatory mutant which apparently plays a very important role in toxoflavin resistance. It is unfortunate that no further characterization was made with respect to its function/role. Since this was the only mutant isolated from the screen of the transcription factor library it could have been characterized more thoroughly.
2. The RNAseq on toxoflavin tolerance yielded a surprisingly large number of loci potentially involved – interesting phenotypic links have been made to fatty acid biosynthesis and resistance to ROS – molecular analysis linking gene expression of loci to the regulator could have made the analysis more complete. In addition finding only one regulatory mutant in the screen is surprising since many loci have a significant change in gene expression. These two sets of data could have been linked.
3. Have the PCR products as evidence for the presence of the bacteria and fungus been sequenced?
4. Enhancement of spore production by *B. glumae* appears clear (via chemical signaling) but no evidence of mechanism is provided.
5. Based on the results it is postulated that *B. glumae* is carried on *Fusarium* spores. This is an interesting observation which has important implications in the disease. What are spore sizes and of *B. glumae* cells? Is this physical interaction sterically possible in terms of size and weight? Microscopy studies would have considerably added evidence to this hypothesis. Are there other examples of bacterial cells carried in spores? Do not really understand how can *B. glumae* be more resistant to UV if it resides on spores.
6. The manuscript requires editing: some examples listed here below
 - a. Line 14 – the word 'likely' is not appropriate for an abstract of a high impact journal
 - b. Line 26-28 – sentence is not clear
 - c. Line 42 – do not use 'one' but 'firstly'
 - d. Line 43 – do not use word 'two' but 'secondly'
 - e. Line 86 – basis not bases

Reviewer #2 (Remarks to the Author):

Authors present their data of interactions between the rice pathogens, the bacterium *Burkholderia glumae* and the fungus *Fusarium graminearum*. Because Fg was shown to be resistant to the Bg toxin toxoflavin in an earlier study, the authors wonder if this property might underlie an ecological interaction between the two microbes. They find that co-cultivation induces sporulation of the fungus w/ bacterial attachment protecting from UV and several other microbial attributes including virulence and dispersal. To determine how Fg is resistant to toxoflavin, they screen an already created transcription factor knock out library and find 1 (only 1? It was not clear) mutant out of 657 tf ko strains that is sensitive to toxoflavin. They also present data to suggest resistance to toxoflavin is associated with lipid metabolism in Fg.

Major concerns

1. TF FGSG_07589. The TF mutant is not complemented. This is critical missing piece of data in this manuscript. The mutant strain must be complemented (southern, not PCR, shown of the complement) and shown to restore WT resistance to toxoflavin. Also, is there more information one can give about TF FGSG_07589? E.g. such as homologs in other fungi? No information is presented if this TF shows any similarity to known TFs.
2. The authors relate lipids and sensitivity to ROS in this manuscript in a confusing manner. More clarity is required. As they have an assay for screening mutants, it would really help strengthen the current tenuous connection with known ROS defense mutants or make a lipid biosynthesis mutants. In particular there are many ROS mutants available in Fg:
 - a. (Fgap1, <https://www.ncbi.nlm.nih.gov/pubmed/26656279>, <https://www.ncbi.nlm.nih.gov/pubmed/24349499>)
 - b. FgSKN7 and FgATF1 (<https://www.ncbi.nlm.nih.gov/pubmed/25040476>)
 - c. Sod1 (<https://www.ncbi.nlm.nih.gov/pubmed/27037138>)
 - d. ELP3 (<https://www.ncbi.nlm.nih.gov/pubmed/25083910>)
 - e. And many more
3. Toxoflavin induces ROS apparently, where? In fungus? In bacterium? In plant?
4. Can we see Nile red staining of all three fungi after treatment with olive oil? The finding that olive oil protects Fg but not *Magnaporthe* or *Colletotrichum* to toxoflavin weakens the hypothesis that lipid droplets are involved.
5. Authors suggest that "ROS produced by toxoflavin might induce a hyper-oxidative state in GZ03639, leading to developmental differentiation and enhanced spore-mediated dispersion." This can be examined by physiological testing with H₂O₂, menadione, etc.
6. Authors say "Although trichothecene production is not essential for initial infection of Fg, it increases systemic colonisation in spikelets²⁷". This is a wheat ms, is this true for rice as well?

Minor

1. Assume you used olive oil due to percent oleic acid in it? Can you give average?
2. Typos in text, e.g. line 45 the reference was not incorporated into the references. This occurred several times in text.
3. Perhaps tone down the airplane analogies (e.g. luxury passenger plane)

Reviewer #3 (Remarks to the Author):

Comments

The topic is very interesting. However, there are several critical points:

- 1) In general: although the data are interesting, there are many speculations which are not

supported by real data.

- 2) The figures are all too small, and they therefore do not support the data outlined in the text, e.g. the plate assays Fig. 1a, or Fig. 1b: what should be seen in Fig. 1b?
 - 3) The "nested plate system" is badly explained
 - 4) The same is with "capillary tube"
 - 5) P.3 ll37 : The numbers in the text don't fit to the numbers in the Figure 1.
 - 6) Which genes were used to amplify specific band in the PCR approach? The F.g-specific PCR bands are very fade even in the control. This PCR is not convincing
 - 7) P.3, ll 40: it is not experimentally proven that the co-existence of both pathogens results in resistance against toxoflavin
 - 8) The authors raised the question "what is the mechanism by which Fg develops resistance to toxoflavin produced by Bg?". However, identification of a TF which seems to be involved is not the explanation for the mechanism. The authors could test a toxin-free bacterium together with a sensitive F.g. strain in a virulence assay.
 - 9) Fig. 2b nothing to see
 - 10) Fig. 2c: this staining is not a proof for lipid accumulation. Nile blue stains also membranses. Therefore, the structures could be also be kind of vesicles, e.g. vacuoles. Nils blue is also used for staining of proteins in SDS gels. A quantification of lipids is needed
 - 11) Fig. 2d Magnaporthe doesn't grow already on MM, therefore this plate assay doesn't say much.
 - 12) P5 ll79: "These findings indicate that the formation of lipid droplets in response to exogenous toxins is an important detoxification mechanism of Fg ..." However, the lipid accumulation has not been proven by quantification of lipids. The staining is only a first indication, but not a proof
 - 13) P6: to confirm the role of ROS, the authors should use catalase-containing media
 - 14) Did the authors test more Fg strains? Are the effects the same ?
 - 15) Ll 109: " The presence of Bg may enhance trichothecene production by Fg, which aids in disease progression" . Speculation. Why not determine trichothecene content in planta? There are also TRI-free strains available
 - 16) Ll 142 "Lipid-mediated detoxification of the Bg-produced antibiotic, toxoflavin, allows Fg to tolerate the bacterium." Again, it is a speculation only based on a staining
 - 17) p7 117ff: „actively and specifically moves forward“: not proven
 - 18) ll 147: "chemically attracted" , again speculation. They authors did not show that there is a chemical principle to attract the bacterium. It could be e.g. a peptide which could be easily demonstrated by boiling the culture filtrate
- Altogether, these are first interesting results, but there are too many speculations which need experimental proof.

Reviewer #1 (Remarks to the Author):

Manuscript entitled 'Cooperative interactions between seed-borne bacterial and air-borne fungal pathogens on rice'

This manuscript presents studies on the presence and role of a fungal (*Fusarium graminearum*) and bacterial pathogen (*Burkholderia glumae*) on a rice grain disease. The two pathogens were found to co-exist in the wild in rice grains and in causing a more severe/aggressive disease when co-inoculated. Other studies indicate that the two microorganisms can interact, stimulate production of toxins and the bacterium can live on the fungal spores. A major conclusion is that the fungus and bacterium can co-exist and cooperate.

General comment

This study presents some interesting data on the interaction between *Fusarium graminearum* and *Burkholderia glumae*. Some of the data is solid whereas some is less and is at its early stages being rather preliminary. Studies deciphering interactions between fungi and bacteria are at large unknown, therefore this is a relatively novel field. Overall this study has some interesting and novel results but it lacks focus presenting in several cases preliminary data from several angles. See specific comments below.

1. The screening of *Fusarium* mutants resistant to toxoflavin yielded one mutant, a putative regulatory mutant which apparently plays a very important role in toxoflavin resistance. It is unfortunate that no further characterization was made with respect to its function/role. Since this was the only mutant isolated from the screen of the transcription factor library it could have been characterized more thoroughly.

>>> We appreciate the reviewer's feedback. We did identify a total of three transcription factor deletion mutants, $\Delta GzZC190$, $\Delta GzbZIP005$, and $\Delta GzC2H008$, which all exhibited sensitivity to toxoflavin (Supplementary Fig. 1). The mutant, $\Delta GzZC190$, was more extensively studied since it exhibited the greatest sensitivity to toxoflavin and $GzZC190$ was the only gene that was up-regulated in the wild-type strain following toxoflavin treatment (Supplementary Table 1). However, all three mutants had 17 phenotypes evaluated, including vegetative growth, sexual development, conidiation, virulence, toxin production, and stress responses, among others. Compared with the wild-type strain, the phenotypes examined remained unchanged in the $\Delta GzZC190$ and $\Delta GzbZIP005$ strains. In contrast, the $\Delta GzC2H008$ mutant exhibited pleiotropic changes in phenotype, including abnormal sexual development, loss of virulence, and reduced production of toxins. This information is provided in our revised manuscript. Our revised manuscript also includes RNA-seq analyses of $\Delta GzbZIP005$ and $\Delta GzC2H008$, and data regarding the triacylglyceride content of the wild-type and $\Delta GzZC190$ strains in the presence and absence of toxoflavin (Fig. 3 and Supplementary Fig. 7). After toxoflavin treatment, there were no significant differences in the total TAGs between the WT and $\Delta GzZC190$ strains. A similar profile was observed for the amounts of TAGs containing linolenic acid, oleic acid, or linoleic acid (Fig. 3a). Interestingly, linolenic acid levels were reduced in the toxoflavin-treated $\Delta GzZC190$ strain compared with the toxoflavin-treated WT strain ($P < 0.05$) and the untreated $\Delta GzZC190$ ($P < 0.01$) strain (Fig. 3a, Supplementary Fig. 7). Correspondingly, genes involved in the biosynthesis of linolenic acid were found to be up-regulated in the WT strain compared to the $\Delta GzZC190$ strain after

toxoflavin treatment, whereas most genes were down-regulated in the toxoflavin-treated $\Delta GzZC190$ strain compared to the untreated $\Delta GzZC190$ strain (Fig. 3b). These results are consistent with the observed changes in the TAGs containing linolenic acid in response to toxoflavin.

2. The RNAseq on toxoflavin tolerance yielded a surprisingly large number of loci potentially involved – interesting phenotypic links have been made to fatty acid biosynthesis and resistance to ROS – molecular analysis linking gene expression of loci to the regulator could have made the analysis more complete. In addition finding only one regulatory mutant in the screen is surprising since many loci have a significant change in gene expression. These two sets of data could have been linked.

>>> Thank you for this feedback. In addition to $\Delta GzZC190$, mutants $\Delta GzbZIP005$ and $\Delta GzC2H008$ also exhibited sensitivity to toxoflavin, although their sensitivity was less than that of $\Delta GzZC190$. To characterize these three mutants and identify possible links among their genes in relation to toxoflavin resistance, we obtained transcriptome data for the wild-type strain and the three mutant strains in the presence and absence of toxoflavin for our revised manuscript. Under the toxoflavin treatment conditions, 2278, 2117, and 1460 differentially expressed genes (DEGs) were identified for $\Delta GzZC190$, $\Delta GzC2H008$, and $\Delta GzbZIP005$, respectively, compared with the wild-type strain. Interestingly, many of the DEGs were significantly enriched in several pathways related to fatty acids (Supplementary Fig. 5 and Supplementary Tables 2-4), including fatty acid degradation (fgr00071), fatty acid biosynthesis (fgr01212), and fatty acid metabolism (fgr00061). However, these pathways did not have similar profiles in the three mutants. For example, gene expression associated with fatty acid degradation was down-regulated in the mutant $\Delta GzZC190$ compared to the wild-type strain, yet the same gene expression profile was up-regulated in $\Delta GzC2H008$. But, gene expression associated with fatty acid metabolism was up-regulated in $\Delta GzC2H008$ and $\Delta GzbZIP005$. These data suggest that *F. graminearum* does not undergo quantitative changes in fatty acid content in the presence of toxoflavin, yet internal metabolism of fatty acids does occur and is needed for a toxoflavin response. Indeed, we observed that genes associated with fatty acids containing α -linolenic acid were up-regulated in the wild-type strain compared to the mutant TF strains in response to toxoflavin (Fig. 3).

3. Have the PCR products as evidence for the presence of the bacteria and fungus been sequenced?

>>> Yes, we sequenced the PCR products to confirm the identities of the bacterium and fungus species.

4. Enhancement of spore production by *B. glumae* appears clear (via chemical signaling) but no evidence of mechanism is provided.

>>> We did observe that *Bg* grown in culture induced spore production by *Fusarium graminearum* (Fg), while greater spore production was induced by toxoflavin, and to a lesser extent by H_2O_2 (Fig. 4 and Supplementary Fig. 10). When we analyzed our RNA-seq data from the WT strain following toxoflavin treatment for genes related to spore production in Fg, 24 up-regulated genes were identified, including genes that encode transcription factors (TFs)

(Supplementary Table 5). When five of these genes were individually disrupted, spore formation was reduced in each of the corresponding mutants. One of these genes encodes the TF, FGSG 01877. This TF was up-regulated in response to toxoflavin in the present study, and in a previous study, overexpression of this TF resulted in enhanced spore production (Jung et al., 2014). Thus, the production of toxoflavin by Bg appears to induce several TFs that are involved in the induction and/or process of spore formation in Fg.

5. Based on the results it is postulated that *B. glumae* is carried on *Fusarium* spores. This is an interesting observation which has important implications in the disease. What are spore sizes and of *B. glumae* cells? Is this physical interaction sterically possible in terms of size and weight? Microscopy studies would have considerably added evidence to this hypothesis. Are there other examples of bacterial cells carried in spores? Do not really understand how can *B. glumae* be more resistant to UV if it resides on spores.

>>> The spore size of Fg is approximately $3\sim 5 \times 25\sim 50 \mu\text{m}$ and the size of a Bg cell is $0.7 \times 1.5\sim 2.5 \mu\text{m}$. In our revised manuscript we have added microscopy images of Bg cells physically attached to Fg spores. While bacterial cells often attach to fungal hyphae and are carried during fungal hyphal growth in soil systems, we could not find other examples of bacterial cells carried in fungal spores through air. Regarding UV protection, we hypothesize that the Fg spores act as a type of umbrella for the Bg cells to block UV exposure. As shown in Supplementary Fig. 12 and Supplementary Table 6, Fg spores were found to be more resistant to UV than Bg cells.

6. The manuscript requires editing: some examples listed here below

- a. Line 14 – the word ‘likely’ is not appropriate for an abstract of a high impact journal
- b. Line 26-28 – sentence is not clear
- c. Line 42 – do not use ‘one’ but ‘firstly’
- d. Line 43 – do not use word ‘two’ but ‘secondly’
- e. Line 86 – basis not bases

>>> We appreciate this feedback. We reviewed all of the suggestions and also had our entire manuscript reviewed for grammar by a native English speaking scientific copyeditor.

Reviewer #2 (Remarks to the Author):

Authors present their data of interactions between the rice pathogens, the bacterium *Burkholderia glumae* and the fungus *Fusarium graminearum*. Because Fg was shown to be resistant to the Bg toxin toxoflavin in an earlier study, the authors wonder if this property might underlie an ecological interaction between the two microbes. They find that co-cultivation induces sporulation of the fungus w/ bacterial attachment protecting from UV and several other microbial attributes including virulence and dispersal. To determine how Fg is resistant to toxoflavin, they screen an already created transcription factor knock out library and find 1 (only 1? It was not clear) mutant out of 657 tf ko strains that is sensitive to

toxoflavin. They also present data to suggest resistance to toxoflavin is associated with lipid metabolism in Fg.

Major concerns

1. TF FGSG_07589. The TF mutant is not complemented. This is critical missing piece of data in this manuscript. The mutant strain must be complemented (southern, not PCR, shown of the complement) and shown to restore WT resistance to toxoflavin. Also, is there more information one can give about TF FGSG_07589? E.g. such as homologs in other fungi? No information is presented if this TF shows any similarity to known TFs.

>>> Thank you for this feedback. Accordingly, we generated a complemented strain that represents the reintroduction of FGSG_07589 into the deletion mutant, $\Delta GzZC190$. Reintroduction was confirmed by Southern analysis as recommended, and FGSG_07589 transcripts were quantified by real-time PCR (Supplementary Fig. 2). Sensitivity to toxoflavin was restored in the complemented strain to the level of the wild-type strain (Fig. 2). We further provide a phylogenetic tree for FGSG_07589 and two other genes in our revised manuscript (FGSG_02939, and FGSG_01106 genes; Supplementary Fig. 4). We found homologs of FGSG_07589 are conserved in the Fusarium genus, while FGSG_07589 is unique to Fg (Supplementary Fig. 4).

2. The authors relate lipids and sensitivity to ROS in this manuscript in a confusing manner. More clarity is required. As they have an assay for screening mutants, it would really help strengthen the current tenuous connection with known ROS defense mutants or make a lipid biosynthesis mutants. In particular there are many ROS mutants available in Fg:

- a. Fgap1, (<https://www.ncbi.nlm.nih.gov/pubmed/26656279>, <https://www.ncbi.nlm.nih.gov/pubmed/24349499>)
- b. FgSKN7 and FgATF1 (<https://www.ncbi.nlm.nih.gov/pubmed/25040476>)
- c. Sod1 (<https://www.ncbi.nlm.nih.gov/pubmed/27037138>)
- d. ELP3 (<https://www.ncbi.nlm.nih.gov/pubmed/25083910>)
- e. And many more

>>> Thank you for your feedback and your time in providing specific resources. We have now generated mutants carrying individual deletions of FgSKN7, FgATF1, Sod1, and ELP3 from the wild-type strain, GZ03639. The sensitivity of these mutants to H₂O₂ and toxoflavin were tested (Supplementary Fig. 7). Only the *sod1* deletion mutant was significantly inhibited by toxoflavin, and this sensitivity was not restored with catalase supplementation. In addition, the transcript levels of five *sod* genes in Fg WT and $\Delta GzZC190$ were detected. All five genes were significantly down-regulated in $\Delta GzZC190$ after exposure to toxoflavin, yet they were up-regulated in the WT strain under the same conditions (Supplementary Fig. 8). Thus, it appears that rapid scavenging of superoxides generated by toxoflavin is a crucial mechanism employed by Fg for toxoflavin resistance.

3. Toxoflavin induces ROS apparently, where? In fungus? In bacterium? In plant?

>>> Toxoflavin has been shown to induce ROS in fungi and plants (Karki et al., 2012).

4. Can we see Nile red staining of all three fungi after treatment with olive oil? The finding that olive oil protects Fg but not Magnaporthe or Colletotrichum to toxoflavin weakens the hypothesis that lipid droplets are involved.

>>> Thank you for this valuable feedback. We performed Nile red staining of all three fungi following their treatment with oleic acid, linoleic acid, and linolenic acid. Treatment resulted in the formation of larger lipid droplets in Fg, while the lipid droplets in Mo and Cg appeared unaffected (Fig. 2 and Supplementary Fig. 6).

5. Authors suggest that “ROS produced by toxoflavin might induce a hyper-oxidative state in GZ03639, leading to developmental differentiation and enhanced spore-mediated dispersion.” This can be examined by physiological testing with H₂O₂, menadione, etc.

>>> Thank you for your suggestion. We examined spore production in the presence of 0.5 mM H₂O₂ and a significant enhancement in spore production was observed (Supplementary Fig. 10). In addition, RNA-Seq data showed that 24 genes particularly related to conidiation were up-regulated after toxoflavin treatment. Taken together, these results suggest that ROS can enhance the expression of genes related to conidiation and lead to increased spore production.

6. Authors say “Although trichothecene production is not essential for initial infection of Fg, it increases systemic colonisation in spikelets²⁷ “. This is a wheat ms, is this true for rice as well?

>>> We appreciate your point, and we have revised our sentence as follows: “While DON production is not essential for the initial infection of Fg in wheat plants, yet it helps systemic colonisation in wheat spikelets⁴⁵, coinoculation of Fg and Bg resulted in greater disease severity and higher DON production in infected rice heads (Fig. 6 and Supplementary Fig. 11).”

Minor

1. Assume you used olive olive due to percent oleic acid in it? Can you give average?

>>> We repeated all of our experiments with 1% sodium oleate. Based on the quantification of triacylglycerides, we also tested the effects of linoleic acid and linolenic acid.

2. Typos in text, e.g. line 45 the reference was not incorporated into the references. This occurred several times in text.

>>> We apologize for this error. We have now rechecked that all of the references cited in our revised manuscript are included in our references list.

3. Perhaps tone down the airplane analogies (e.g luxury passenger plane)

>>> Thank you for this feedback. We have rearranged the text in the paragraph that originally contained this analogy. As part of these revisions, the following text is now

included: “In contrast, the attachment of bacteria to fungal hyphae provides a more rapid and expansive route of dispersal.”

Reviewer #3 (Remarks to the Author):

Comments

The topic is very interesting. However, there are several critical points:

1) In general: although the data are interesting, there are many speculations which are not supported by real data.

>>> We would respectfully submit that we have addressed the “speculations” you refer to with the addition of data we now include in our revised manuscript as a result of the reviewers’ feedback. These data include the identification of two additional transcription factor deletion mutants, $\Delta GzbZIP005$ and $\Delta GzC2H008$, which exhibit sensitivity to toxoflavin, albeit to a lesser extent than $\Delta GzZC190$ (Supplementary Fig. 1). In addition, 17 phenotypes were examined for all three mutants, including vegetative growth, sexual development, conidiation, virulence, toxin production, and stress responses, among other. These phenotypes were unchanged for $\Delta GzZC190$ and $\Delta GzbZIP005$ compared with the wild type strain. In contrast, $\Delta GzC2H008$ exhibited pleiotropic phenotype changes that included abnormal sexual development, loss of virulence, and reduced toxin production. We additionally performed RNA-seq analyses of $\Delta GzbZIP005$ and $\Delta GzC2H008$ and have added these data into our revised manuscript. To further characterize $\Delta GzZC190$, triacylglyceride content was analyzed in the wild-type and $\Delta GzZC190$ strains with and without toxoflavin treatment (Fig.3 and Supplementary Fig. 7). We have also revised the Discussion of our manuscript to more clearly present these new results in relation to our original data.

2) The figures are all too small, and they therefore do not support the data outlined in the text, e.g. the plate assays Fig. 1a, or Fig. 1b: what should be seen in Fig. 1b?

>>> Thank you for this feedback. We have reorganized our figures and closely reviewed their content, as well as the corresponding references in our revised manuscript.

3) The “nested plate system” is badly explained

>>> Thank you for this feedback. We now provide further details regarding the nested plate system in the Methods section of our revised manuscript.

4) The same is with “capillary tube”

>>> Thank you for this feedback. We also provide additional details regarding the capillary tube-based attraction assay in our revised manuscript.

5) P.3 ll37 : The numbers in the text don't fit to the numbers in the Figure 1.

>>> We apologize for this mistake. We have corrected the values reported for Figure 1 in the main text.

6) Which genes were used to amplify specific band in the PCR approach? The F.g-specific PCR bands are very fade even in the control. This PCR is not convincing

>>> *F. graminearum*-specific primers and a *B. glumae* primer set designed for a *rhs* family gene were used to detect Fg and Bg, respectively. The image of the separation gel for the PCR products does not provide the same signal intensity that direction observation of the gel provided. We were able to excise bands of interest, purify them, and have them sequenced. Therefore, sufficient material was obtained from the gel bands.

7) P.3, ll 40: it is not experimentally proven that the co-existence of both pathogens results in resistance against toxoflavin

>>> Thank you for this feedback. We have rewritten the relevant text as follows. "The seed-borne bacterial plant pathogen, *Burkholderia glumae* (Bg), is one of the causal agents of bacterial panicle blight in rice fields. This bacterium produces toxoflavin, which is both a critical virulence factor and an antimicrobial that induces production of superoxides and hydrogen peroxide (H₂O₂). Thus, Bg has the potential to monopolise rice grains by blocking the growth of saprophytic fungi and other pathogens. However, we have observed that the air-borne fungal plant pathogen, *Fusarium graminearum* (Fg), is resistant to toxoflavin. Therefore, we examined the potential for Bg and Fg to interact and co-exist, particularly regarding the mechanism by which Fg develops resistance to toxoflavin that is produced by Bg."

8) The authors raised the question "what is the mechanism by which Fg develops resistance to toxoflavin produced by Bg?" However, identification of a TF which seems to be involved is not the explanation for the mechanism. The authors could test a toxin-free bacterium together with a sensitive F.g. strain in a virulence assay.

>>> Thank you for your feedback and suggestions. We have quantified the triacylglyceride (TAG) content of the WT and $\Delta GzZC190$ strains with and without toxoflavin treatment by LC-MS/MS to further investigate the mechanism involved. The total TAG content of $\Delta GzZC190$ prior to treatment was higher than the total TAG content of the WT strain without toxoflavin treatment. Toxoflavin treatment of $\Delta GzZC190$ then led to a reduction in TAGs containing linolenic acid compared to the WT strain treated with toxoflavin, and $\Delta GzZC190$ without toxoflavin treatment. These results are consistent with the expression profiles we obtained for genes involved in the biosynthesis of linolenic acid in the WT and $\Delta GzZC190$ strains (Fig. 3). Furthermore, these results suggest that TAGs containing linolenic acid are protected within LDs, and the genes involved in the biosynthesis of linolenic acid are induced in response to ROS generated from toxoflavin. Further support for this model was provided by the treatment of $\Delta GzZC190$ with linolenic acid (or oleate) and the subsequent restoration

of toxoflavin sensitivity. Thus, TAGs containing linolenic acid may play a role in resistance to ROS stress that is generated in response to toxoflavin.

9) Fig. 2b nothing to see

>>> Thank you for this feedback. In our revised manuscript we have moved Figure 2b to Supplementary Figure 3 and have reorganized Figure 2.

10) Fig. 2c: this staining is not a proof for lipid accumulation. Nile blue stains also membranes. Therefore, the structures could be also be kind of vesicles, e.g. vacuoles. Nile blue is also used for staining of proteins in SDS gels. A quantification of lipids is needed

>>> Thank you for this feedback and we appreciate your recommendation. We chose to quantify the triacylglyceride content of fungal hyphae in the presence and absence of toxoflavin by LC/MS-MS. The total triacylglyceride content did not significantly differ among the samples examined, with the exception that the levels of TAGs containing linolenic acid were lower in the hyphae following toxoflavin treatment. These data are presented in Figure 3 and Supplementary Figure 7.

11) Fig. 2d Magnaporthe doesn't grow already on MM, therefore this plate assay doesn't say much.

>>> We appreciate the reviewer's feedback. However, our direct observations were that *M. oryzae* grows on the surface of MM without robust aerial hyphae. Therefore, we reimaged the plates to achieve a better representation of the data obtained.

12) P5 l179: "These findings indicate that the formation of lipid droplets in response to exogenous toxins is an important detoxification mechanism of *Fg* ..." However, the lipid accumulation has not been proven by quantification of lipids. The staining is only a first indication, but not a proof

>>> We appreciate the reviewer's feedback. Therefore, we have added LC-MS/MS data that provides quantification of the triacylglyceride content in the wild-type and $\Delta GzZC190$ strains with and without toxoflavin treatment (Fig. 3 and Supplementary Fig. 7).

13) P6: to confirm the role of ROS, the authors should use catalase-containing media

>>> Thank you for this feedback. Accordingly, we performed the relevant experiments with catalase-containing media. As a result, sensitivity to H_2O_2 was successfully restored in the wild-type and $\Delta GzZC190$ strains, yet sensitivity to toxoflavin was not. In addition, we generated deletion mutants of *FgSKN7*, *FgATF1*, *Sod1*, and *ELP3* from the wild-type strain, GZ03639. When these mutants were tested for sensitivity to toxoflavin (Supplementary Figure 8), only the *sod1* deletion mutant was significantly inhibited by toxoflavin. Moreover, toxoflavin sensitivity of this mutant was not restored by catalase. When transcript levels of five *sod* genes were examined in *Fg*, all five genes were found to be down-regulated in the toxoflavin sensitive mutant, $\Delta GzZC190$, compared to the wild-type strain (Supplementary Fig.

9). Based on these results, it appears that rapid scavenging of superoxides generated by toxoflavin is one of the mechanisms mediating toxoflavin resistance.

14) Did the authors test more Fg strains? Are the effects the same?

>>> We performed our experiments with four Fg strains, and the results are shown in Supplementary Fig. 10b.

15) Ll 109: “ The presence of Bg may enhance trichothecene production by Fg, which aids in disease progression” . Speculation. Why not determine trichothecene content in planta? There are also TRI-free strains available

>>> Thank you for this feedback. As suggested, we quantified trichothecene content and disease severity *in planta*. Single inoculations of BGR1 or GZ03639 on rice heads in the late-flowering stage resulted in disease severities of 15% and 33%, respectively. In contrast, co-inoculation of both pathogens resulted in 82% disease severity and disease progression to the stem (Fig. 6a, Supplementary Fig. 11). DON production was also approximately 2-fold higher in the rice plants that were inoculated with both Fg and Bg compared with the rice plants that were only inoculated with Fg (Fig. 6b).

16) Ll 142 “Lipid-mediated detoxification of the Bg-produced antibiotic, toxoflavin, allows Fg to tolerate the bacterium.” Again, it is a speculation only based on a staining

>>> We appreciate the reviewer’s feedback and we have now added LC-MS/MS data to our revised manuscript that provides quantification of the triacylglyceride content in the wild-type and $\Delta GzZC190$ strains with and without toxoflavin treatment (Fig. 3 and Supplementary Fig. 7).

17) p7 117ff: „actively and specifically moves forward“: not proven

>>> We have revised this sentence as follows: “A similar observation was made when a greater number of BGR1 cells were found to have moved into a capillary tube containing a culture filtrate of GZ03639, compared with other capillaries that contained pure medium or a culture filtrate of another rice fungal pathogen, *M. oryzae* (Fig. 5c). Furthermore, the Fg culture filtrates did not attract the DC3000 cells (Fig. 5c).”

18) ll 147: “chemically attracted” , again speculation. They authors did not show that there is a chemical principle to attract the bacterium. It could be e.g. a peptide which could be easily demonstrated by boiling the culture filtrate

>>> We appreciate the recommendation you provide regarding the boiling of the culture filtrate, which also attracted the Bg cells. We changed to “signaling molecules” instead of chemical attraction.

Reviewer #1 (Remarks to the Author):

Manuscript entitled 'Cooperative interactions between seed-borne bacterial and air-borne fungal pathogens on rice'

This manuscript has been extensively reviewed according to the comments of the three reviewers.

In reply to my comments, the authors have now provided data on the three transcription factors which are involved in toxoflavin resistance by performing transcriptome analysis and phenotypic assays. They have also performed microscopy images showing Bg cells attached to Fg spores. Throughout the manuscript attempts have been made to link resistance to toxoflavin to metabolic changes and how BG and Fg are potentially involved in signaling. In addition authors have responded to the comments of the other two reviewers including the important complementation of TF mutant GzZC190.

In summary, the revised version reads better and is more linked and complete. The manuscript contains some solid data and some data which is more speculative with respect to resistance to toxoflavin and BG and Fg communication. The authors however have reported an important and novel interkingdom interaction and have gone at some length in determining possible mechanisms which allow Bg and Fg to co-exist and benefits that this might bring.

Reviewer #2 (Remarks to the Author):

The Authors have met my suggestions adequately. enjoyed reading this.

The reviewers did not raise additional comments that we need to address.

REVIEWERS' COMMENTS:

Reviewer #1 (Remarks to the Author):

Manuscript entitled 'Cooperative interactions between seed-borne bacterial and air-borne fungal pathogens on rice'

This manuscript has been extensively reviewed according to the comments of the three reviewers.

In reply to my comments, the authors have now provided data on the three transcription factors which are involved in toxoflavin resistance by performing transcriptome analysis and phenotypic assays. They have also performed microscopy images showing Bg cells attached to Fg spores. Throughout the manuscript attempts have been made to link resistance to toxoflavin to metabolic changes and how BG and Fg are potentially involved in signaling. In addition authors have responded to the comments of the other two reviewers including the important complementation of TF mutant GzZC190.

In summary, the revised version reads better and is more linked and complete. The manuscript contains some solid data and some data which is more speculative with respect to resistance to toxoflavin and BG and Fg communication. The authors however have reported an important and novel interkingdom interaction and have gone at some length in determining possible mechanisms which allow Bg and Fg to co-exist and benefits that this might bring.

Reviewer #2 (Remarks to the Author):

The Authors have met my suggestions adequately. enjoyed reading this.